# FATE: Focal-modulated Attention Encoder for Multivariate Time-series Forecasting

## Abstract

Accurate multivariate time-series forecasting is crucial for understanding and mitigating the effects of climate change, as reliable long-horizon predictions support effective monitoring and informed decision-making. Existing neural approaches ranging from CNNs and RNNs to attention-based Transformers have achieved notable progress. Yet, they often suffer from two key limitations: difficulty in capturing hierarchical spatiotemporal dependencies and computational inefficiencies when scaling to high-dimensional meteorological data. We propose FATE (Focal-modulated Attention Encoder), a new Transformer architecture tailored for robust multivariate time-series forecasting. FATE introduces a tensorized focal modulation mechanism that enhances spatiotemporal dependency modeling while maintaining scalability. To improve interpretability, we further design dual modulation scores that identify critical environmental features driving the forecasts. Comprehensive experiments on seven diverse real-world datasets including benchmark energy, traffic, and large-scale climate datasets demonstrate that FATE consistently surpasses state-of-the-art methods, particularly on long-horizon and high-variability settings. Extensive ablations confirm the generalization ability of FATE across heterogeneous forecasting tasks. To foster reproducibility and future research, we will release the full implementation.

## 1 Introduction

The Transformer architecture (Vaswani et al., 2017a) has become a cornerstone of modern deep learning, driving breakthroughs in natural language processing (Brown et al., 2020; Radford et al., 2019; Devlin et al., 2018b; Radford et al., 2021), computer vision (Dosovitskiy et al., 2020; Zhu et al., 2021; Yang et al., 2022), and large-scale foundation models (Kaplan et al., 2020). Motivated by this success, recent works have applied Transformers to multivariate time-series forecasting, leveraging their ability to model pairwise dependencies and extract multi-level sequence representations (Wu et al., 2021a; Nie et al., 2023). However, their effectiveness in this domain remains contested. Notably, simple linear models rooted in classical statistics (Box & Jenkins, 1968) have been shown to outperform Transformers in both accuracy and efficiency (Zeng et al., 2023a; Das et al., 2023a). At the same time, emerging architectures that explicitly model multivariate correlations (Zhang & Yan, 2023a; Ekambaram et al., 2023) underscore the limitations of vanilla self-attention for complex time-series dynamics.

We identify three fundamental shortcomings of existing Transformer-based approaches for multivariate forecasting: (1) *Permutation-invariant self-attention* fails to capture *temporal order*, leading to weak modeling of *sequential dynamics*. (2) *Uniform attention across tokens* not only overlooks the *varying significance of climate variables across spatiotemporal scales*, but also leads to *computational inefficiencies* when scaling to *high-dimensional meteorological data*. (3) The architecture lacks an explicit mechanism to model *hierarchical spatiotemporal correlations*, which are crucial for *long-horizon forecasting*.

Unlike FocalNet (Yang et al., 2022), which was designed for spatial representation learning in vision tasks, FATE introduces key innovations tailored for multivariate time-series forecasting:

- **Tensorized Attention Design:** FATE preserves the full 3D tensor structure ($X \in \mathbb{R}^{T \times S \times P}$), maintaining temporal and variable axes explicitly. This enables more effective modeling of long-range dependencies through grouped attention across both time and features.

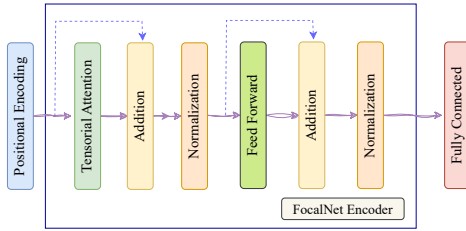 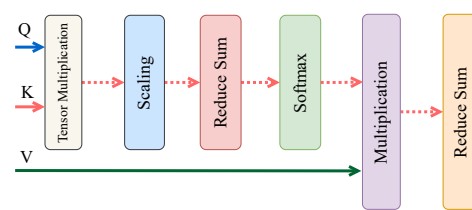

(a) Encoder Architecture of FATE.    (b) Tensorial Focal Modulation

Figure 1: Our proposed architecture consists of two main components: Figure 1a shows the overall architecture of FATE encoder. The input time series data passes first through positional encoding, and then Tensorial Attention, which incorporates spatial as well as temporal information. Figure 1b explains the internal working of the tensorial focal-modulation block. The Query (Q), Key (K), and Value (V) tensors undergo a series of tensor multiplication, scaling, reduction, and softmax operations to create attention maps. These maps are then used by the model to determine which regions of the inputs are more significant.

- **Focal Grouping for Temporal Blocks:** Instead of spatial grids, FATE dynamically defines *temporal focal groups* that adapt to prediction horizons, allowing the model to capture hierarchical temporal dependencies unique to time-series data.

- **Cross-axis Modulation:** Focal modulation is extended beyond temporal steps to the variable dimension, enabling rich cross-feature interactions that are absent in FocalNet.

In this way, FATE is not a simple adaptation of FocalNet, but a principled redesign that leverages the structural properties and forecasting demands of multivariate time-series data.

Long-term variations in temperature, precipitation, wind, and other environmental factors define climate change (Barrett et al., 2015). These shifts have profound global impacts, threatening sustainability in domains such as food security, public health, and energy systems. For instance, a projected increase of up to $2°C$ in global mean temperature this century could severely reduce crop yields. Unlike short-term fluctuations, climate change evolves over decades, driven primarily by greenhouse gas emissions, deforestation, and limited adoption of renewable energy (Latake et al., 2015). Accurate long-horizon forecasting of such multivariate processes is therefore critical. It enables policymakers and practitioners to assess risks, monitor climate drivers, and design mitigation strategies (Huntingford et al., 2019). However, the multidimensional and highly correlated nature of climate data poses significant challenges for existing forecasting models.

To address these challenges, we propose FATE, a novel Transformer that (1) introduces tensorized focal modulation for explicit spatiotemporal correlation modeling, (2) employs dual modulation scores to enhance interpretability, and (3) adaptively emphasizes relevant tokens via selective attention. We evaluate FATE across seven diverse real-world datasets and demonstrate that it consistently outperforms state-of-the-art methods, particularly on long-horizon and high-dimensional climate datasets. Extensive ablation studies further confirm that FATE generalizes effectively across broader multivariate forecasting tasks.

**Contributions.** The main contributions of this work are threefold:

- We introduce FATE, a Transformer architecture with a novel focal-modulation mechanism that preserves 3D tensor structure ($T \times S \times P$) for multivariate time-series forecasting.

- We design *dual modulation scores* that improve both predictive performance and interpretability by identifying critical temporal and variable dependencies.

- We achieve new state-of-the-art results on seven benchmark datasets, including accuracy gains of $13.3\%$, $9.1\%$, and $10.1\%$ on ETTm2 (Zhou et al., 2021a), Weather5k (Han et al., 2024), and LargeST (Liu et al., 2023), respectively, with strong improvements across all other datasets.

## 2 RELATED WORK

**Transformers for Time Series Forecasting.** Transformer architectures (Vaswani et al., 2017a) have achieved remarkable success across NLP (Devlin et al., 2018a; Brown et al., 2020; Radford et al., 2019), computer vision (Dosovitskiy et al., 2021; Bao et al., 2022; He et al., 2021), and speech (Baevski et al., 2020; Hsu et al., 2021) due to their scalability and effective sequence modeling. Vision Transformers (ViTs) divide images into patches to preserve local semantic information (Dosovitskiy et al., 2021; Geiger et al., 2013; Li et al., 2020), while NLP models like BERT (Devlin et al., 2018b) leverage subword tokenization for contextual dependencies. Inspired by these successes, Transformer variants have been widely adapted for time-series forecasting (Jake Grigsby & Qi, 2021; Nie et al., 2023). Early models, such as LogTrans (Li et al., 2019) and Informer (Li et al., 2021), addressed computational inefficiencies via sparse attention. Autoformer (Wu et al., 2021a) introduced decomposition-based inductive biases, FEDformer (Zhou et al., 2022a) employed Fourier-enhanced blocks for seasonal modeling, Pyraformer (Liu et al., 2021) added pyramidal attention for multi-scale dependencies, and Triformer (Cirstea et al., 2022) proposed pseudo-timestamp-based patch attention. Despite these advances, many Transformer forecasters still rely on point-wise or handcrafted attention, limiting their ability to capture semantic relationships across patches or dimensions (Sakaridis et al., 2018; Ashish, 2017; Zhu et al., 2023). For example, Autoformer's fixed auto-correlation modules may fail to generalize, and Triformer does not treat patches as first-class units nor model internal semantics. TimeMixer++ (Wang et al., 2024) advances multi-scale, multi-resolution forecasting by converting time series into 2D time images (via Multi-Resolution Time Imaging, `MRTI`) and separating seasonal/trend components in latent space using dual-axis attention, followed by hierarchical Multi-Scale Mixing (`MCM`) and Multi-Resolution Mixing (`MRM`). This allows parallel modeling of concurrent temporal contexts (daily, weekly, seasonal), improving forecasting, classification, and anomaly detection. TimeTensor (Liang et al., 2024) generalizes linear attention to 3D tensor inputs via Kronecker decomposition, improving efficiency while retaining the standard attention paradigm. In contrast, FATE introduces *tensorized focal modulation*, explicitly preserving 3D spatiotemporal structure, enabling hierarchical and localized context aggregation, and jointly modeling long- and short-range dependencies. This represents a novel architectural strategy distinct from previous tensorized attention mechanisms.

**Self-supervised and Representation Learning.** Transformer adaptations for time series can be categorized into four directions (Kalyan et al., 2021): (i) attention-level modifications for efficiency, (ii) adaptations for stationarity and signal processing, (iii) architectural changes capturing cross-variate and temporal dependencies, and (iv) novel tensor-based designs. Most methods focus on the first three, while few explore fundamental tensor-based redesigns. Self-supervised learning (SSL) has also gained traction for time-series representation learning. Methods such as TNC (Tonekaboni et al., 2021), TS2Vec (Yue et al., 2022), and BTSF (Yang & Hong, 2022) learn rich representations without supervision, whereas Transformer-based SSL models like TST (Zerveas et al., 2021) and TS-TCC (Eldele et al., 2021) remain underexplored for capturing complex temporal and cross-variate dependencies. FATE 's tensorized focal modulation inherently supports richer hierarchical representations, bridging this gap by jointly modeling time, feature, and spatial dimensions. *Focal Modulated Tensorized Encoder* introduces a novel tensorized focal modulation mechanism tailored for multivariate time-series forecasting. It preserves the input's 3D tensor structure ($T \times S \times P$), enables hierarchical spatiotemporal correlation modeling, and applies *tensorized attention design*, *temporal focal grouping*, and *cross-axis modulation*. Unlike prior work, FATE balances efficiency with semantic richness and provides a principled framework for long- and short-range dependency modeling in high-dimensional time series.

## 3 PROPOSED METHODOLOGY

In this section, we present FATE, a *Focal Modulated Tensorized Encoder Transformer* designed for multivariate time-series forecasting. The architecture preserves the full 3D structure of the input tensor to jointly model temporal, spatial (station-wise), and feature dimensions. Central to FATE are tensorized focal modulation mechanisms that efficiently capture hierarchical temporal patterns, cross-station interactions, and feature dependencies, while providing interpretable modulation scores that highlight the contribution of each station and attention head. The following subsections detail

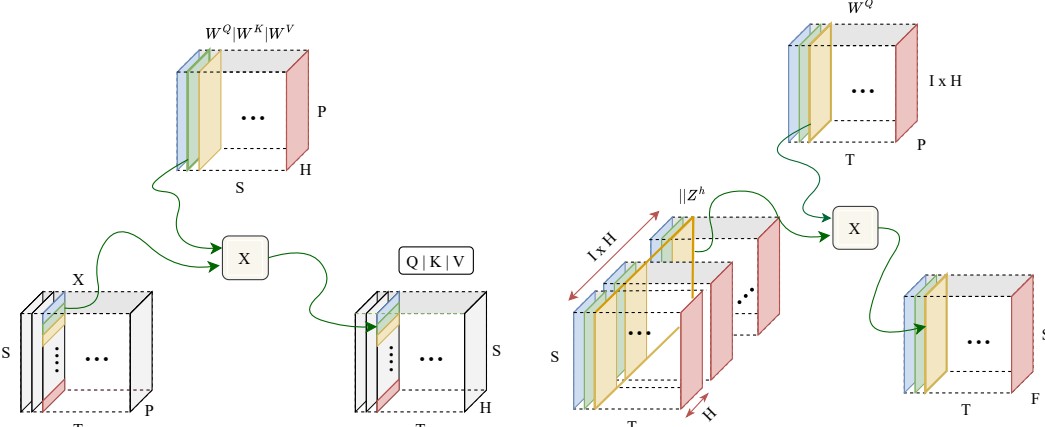

(a) Slice-to-QKV projection: each colored slice of the input tensor $X$ is multiplied by the corresponding weight slice to form the query ($Q$), key ($K$), and value ($V$) tensors.

(b) Multi-head output assembly: the concatenated self-attention outputs are slice-multiplied by weight slices to yield each slice of the output tensor $Y$.

Figure 2: (a) Slice multiplication for QKV extraction. (b) Slice multiplication for multi-head attention output.

the encoder design, the tensorial focal modulation computations, and the aggregation strategy for interpretable predictions.

## 3.1 MULTI-DIMENSIONAL TENSORED FOCALNET ENCODER

We extend the `FocalNet` Transformer (Yang et al., 2022) to propose the *Tensorized Focal Encoder Transformer*, specifically designed to capture complex patterns in multi-dimensional time-series data. Our model operates on climate parameters organized as a 3D tensor $X \in \mathbb{R}^{T \times S \times P}$, where $T$ denotes the temporal dimension, $S$ indexes different stations, and $P$ represents diverse climate parameters (e.g., temperature, humidity, wind speed). The full 3D structure preserves variable–time step relationships and supports parallel yet separate attention across temporal and feature dimensions.

The architecture is encoder-only, as illustrated in Figure 1, and comprises: (i) a positional encoding layer, (ii) a tensorial focal modulation encoder layer, and (iii) a linearly activated fully-connected layer. Each encoder layer integrates tensorial modulation (Sections 3.2 and 3.3) followed by a residual connection and normalization. A densely connected `FFN`, consisting of two linear transformations with `ReLU` activation, follows the modulation layer, and is again succeeded by residual connection and normalization, consistent with (Yang et al., 2022).

## 3.2 TENSORIAL FOCAL MODULATION

To encode temporal hierarchies, we apply a constant positional encoding (Yang et al., 2022) along the time axis $T$ and parameter axis $P$:

$$\text{PE}(\text{pos}, 2i) = \sin\left(\frac{\text{pos}}{10000 \cdot 2^i / P}\right), \tag{1}$$

where pos indexes time and $i$ indexes parameters; the station axis $S$ transmits the encoded values.

Focal modulation replaces pairwise attention with hierarchical context aggregation (Yang et al., 2022), offering three key benefits: (i) improved computational efficiency, (ii) preservation of locality biases, and (iii) non-quadratic long-range dependency modeling. For multivariate time series, FATE leverages this through: (1) nested focal windows that hierarchically aggregate temporal information, and (2) dynamic contextual gating that adapts to input distributions, outperforming fixed receptive fields or conventional attention kernels.

We formalize tensor slices as follows: for a tensor $N \in \mathbb{R}^{X \times Y \times Z}$, $(N_{y,z})_x \in \mathbb{R}^{Y \times Z}$ denotes the $x$-slice, and $(N_z)_{x,y} \in \mathbb{R}^Z$ denotes the $x, y$-slice. Lowercase letters indicate slice sizes.

Tensorial focal modulation operates on $X \in \mathbb{R}^{T \times S \times P}$. We first compute 3D Query $(Q)$, Key $(K)$, and Value $(V)$ tensors, $Q, K, V \in \mathbb{R}^{T \times S \times H}$, via element-wise multiplication with learnable weight tensors $W^Q, W^K, W^V \in \mathbb{R}^{S \times F \times H}$:

$$
\begin{aligned}
(Q_h)_{t,s} &= (X_p)_{t,s} \times (W^Q)_{p,h,s}, \\
(K_h)_{t,s} &= (X_p)_{t,s} \times (W^K)_{p,h,s}, \\
(V_h)_{t,s} &= (X_p)_{t,s} \times (W^V)_{p,h,s}, \quad \forall t = 1..T, s = 1..S.
\end{aligned}
\tag{2}
$$

Next, we compute the multiplicative interaction across time steps:

$$
(\widetilde{R}_{s,s^l})_{t,t^l} = (Q_{s,h})_t \times ((K_{s^l,h})_{t^l})^T, \quad R = \frac{1}{\sqrt{H}} \sum_{s^l=1}^{S} (\widetilde{R}_{t,t^l,s})_{s^l},
\tag{3}
$$

followed by a softmax across the station dimension to obtain attention weights $\widetilde{A} \in \mathbb{R}^{T \times T^l \times S}$:

$$
(\widetilde{A}_s)_{t,t^l} = \text{Softmax}\left((R_{t,t^l,s})_s\right), \quad \forall t, t^l = 1..T.
\tag{4}
$$

Finally, the output $Z \in \mathbb{R}^{T \times C \times D}$ is computed by broadcasting $(\widetilde{A}_s)_{t,t'}$ to match the shape of $(V_{s,d})_{t'}$ and summing over the temporal dimension:

$$
(Z_{s,d})_t = \sum_{t'=1}^{T} \text{broadcast}((\widetilde{A}_s)_{t,t'}) \circ (V_{s,d})_{t'}, \quad \forall t = 1..T.
\tag{5}
$$

### 3.3 Focal Modulation Aggregation

Modulation weights have been widely used for feature selection and interpretability (Wiegreffe & Pinter, 2019). In FATE, the focal modulation tensors $\widetilde{A}$ (Eq. 4) serve to provide interpretable insights into model predictions.

To quantify the relationship between attention heads and cities (stations), we compute *head-wise focal modulation scores*:

$$
N\widetilde{A}_s^h = \sum_{t=1}^{T} \sum_{t'=1}^{T'} A_{t,t',c}^h, \quad \forall h = 1..H, \ c = 1..C.
\tag{6}
$$

We then aggregate across all heads to obtain *city-wise modulation scores*, reflecting the overall contribution of each city to the prediction:

$$
N\widetilde{A}_s = \sum_{h=1}^{H} N\widetilde{A}_s^h, \quad \forall c = 1..C.
\tag{7}
$$

This aggregation completes the tensorial focal modulation process, explicitly linking attention heads to cities and highlighting the importance of each city in driving the model's forecasts.

## 4 Experiments

To rigorously evaluate FATE, we conduct extensive experiments on seven diverse real-world datasets spanning environmental and infrastructural domains, comparing against 17 state-of-the-art baselines including Transformer-, RNN/CNN-, Linear-, and spatial-temporal models. We analyze predictive

performance across short- and long-horizon forecasts using standard metrics (MAE, MSE), benchmark computational and memory efficiency, and provide interpretability through focal modulation visualization. These studies demonstrate FATE's superior accuracy, robustness, and capacity to model multi-scale temporal and spatiotemporal dependencies.

## 4.1 Datasets

We evaluate FATE on seven diverse real-world multivariate time-series datasets, encompassing both environmental (Weather5k, USA-Canada, Europe) and infrastructural (ETTh1, ETTm2, Traffic, LargeST) domains.

**ETTh1** Zhou et al. (2021a) and **ETTm2** Zhou et al. (2021a) are electricity transformer datasets at hourly and minute resolutions, respectively, capturing seasonal and trend-driven consumption patterns. **Traffic** Zhao (2019) consists of road occupancy rates from multiple sensors, serving as a standard benchmark for traffic flow prediction. **Weather5k** Han et al. (2024) is a large-scale dataset with 10 years of hourly measurements from 5,672 weather stations worldwide, including temperature, humidity, wind speed, and other climate parameters. **USA-Canada** Meteorological Development Laboratory, Office of Science and Technology, National Weather Service, NOAA, U.S. Department of Commerce (1987) contains hourly meteorological data from 30 cities (Oct 2012–Nov 2017), enriched with spatial coordinates and temporal features such as hour and day-of-year. The **Europe** dataset Huber et al. (2022) spans 18 European cities (May 2005–Apr 2020), with normalized temporal and meteorological features; the test split covers 2017–2020, and the training/validation span 2005–2017. Finally, **LargeST** Liu et al. (2023) provides traffic data from 8,600 sensors in California over 5 years, including rich sensor metadata for enhanced interpretability.

Across all datasets, FATE consistently outperforms baselines—including Transformer Vaswani et al. (2017b); Yang et al. (2022), 3D-CNN Mehrkanoon (2019b), LSTM Hochreiter & Schmidhuber (1997), and ConvLSTM Shi et al. (2015)—achieving the lowest Mean Absolute Error (MAE) and Mean Squared Error (MSE), particularly on long-horizon and high-dimensional climate datasets.

## 4.2 Additional Implementation Details

**Computational and Memory Requirements.** We use a fixed 30-day input window; for climate datasets, we consider 7 meteorological features, while feature selection for other datasets follows the original data schema. Experiments were conducted on an NVIDIA A100 GPU with 40GB VRAM. Optimizers were selected per architecture following prior best practices.

We analyze FATE 's computational complexity and provide empirical runtime benchmarks against Transformer and CNN-based baselines. While tensorized focal modulation introduces moderate overhead compared to standard Transformers, the performance gains in long-horizon forecasting justify this cost. Preserving the 3D tensor increases memory complexity due to grouped modulation, but efficient projections keep runtime and GPU usage comparable to baseline Transformers.

**Hyperparameters.** Table 1 details all training hyperparameters. Multi-head attention is used in both FATE and Transformer models, with FATE employing four focal levels and eight attention heads to capture hierarchical temporal dependencies. 3D-CNN Mehrkanoon (2019b) and ConvLSTM Shi et al. (2015) models use convolutional layers with kernel sizes tuned for spatiotemporal patterns. LSTM Hochreiter & Schmidhuber (1997) and ConvLSTM models employ recurrent units with hidden dimensions optimized for sequential modeling. Scheduled learning rate decay is applied in FATE and Transformer models, while 3D-CNN, LSTM, and ConvLSTM use fixed rates. Batch sizes are scaled for memory efficiency and stable training.

Table 1: Hyperparameters used for all the models. All hyperparameters were selected using 5-fold cross-validation. Tuning was done independently on each dataset to avoid overfitting or unfair transfer of settings.

| Hyper-parameter | FATE | Transformer | 3D CNN | LSTM | ConvLSTM |
|---|---|---|---|---|---|
| Focal Levels | 4 | 3 | - | - | - |
| Layer Number | 1 | 1 | - | 1 | 3 |
| Head | 8 | 1 | - | - | - |
| Key Dim | 32 | 32 | - | - | - |
| Dense Units | 64 | 64 | 128 | - | - |
| Filters | - | - | 10 | - | 16 |
| Kernel Size | - | - | 4 | - | 13 |
| Hidden Units | - | - | - | 128 | - |
| Learning Rate | Schedule | Schedule | $10^{-4}$ | $10^{-4}$ | $10^{-4}$ |
| Batch Size | 64 | 32 | 128 | 256 | 128 |

## 4.3 Forecasting Results

We evaluate FATE across diverse real-world datasets and benchmark it against 17 state-of-the-art models spanning four categories: (1) *Transformer-based*: iTransformer Nie et al. (2024), Autoformer Wu et al. (2021b), etc.; (2) *RNN/CNN-based*: LSTM Hochreiter & Schmidhuber (1997), ConvLSTM Shi et al. (2015), 3D-CNN Mehrkanoon (2019b); (3) *Linear-based*: DLinear Zeng et al. (2023b), TiDE Das et al. (2023b); (4) *Spatial-temporal* (LargeST dataset): DGCRN Li et al. (2023a), D2STGNN Shao et al. (2022).

Table 2: The test results for temperature prediction, evaluated using the Mean Absolute Error (MAE) and Mean Squared Error (MSE), were obtained for the **USA-Canada** and **Europe** datasets. The best-performing results are highlighted in **bold**, while the second-best are marked in red for clarity.

| Station | Model | MAE 4 hrs | 8 hrs | 12 hrs | 16 hrs | MSE 4 hrs | 8 hrs | 12 hrs | 16 hrs | Station | Model | MAE 3 days | 5 days | 7 days | MSE 3 days | 5 days | 7 days |
|---|---|---|---|---|---|---|---|---|---|---|---|---|---|---|---|---|---|
| Vancouver | Transformer Vaswani et al. (2017b) | 1.238 | 1.858 | 1.987 | 2.146 | 2.566 | 5.787 | 6.617 | 7.748 | Barcelona | Transformer Vaswani et al. (2017b) | 2.608 | 2.901 | 3.347 | 11.702 | 14.660 | 15.926 |
| | 3D CNN Mehrkanoon (2019a) | 1.499 | 1.896 | 2.131 | 2.329 | 3.704 | 5.950 | 7.455 | 8.879 | | 3D CNN Mehrkanoon (2019a) | 2.502 | 3.015 | 3.059 | 10.73 | 13.654 | 15.740 |
| | LSTM Hochreiter & Schmidhuber (1997) | 1.311 | 1.834 | 2.039 | 2.210 | 2.917 | 5.712 | 6.970 | 8.237 | | LSTM Hochreiter & Schmidhuber (1997) | 2.303 | 2.801 | 2.931 | 9.354 | 11.328 | 14.931 |
| | ConvLSTM Shi et al. (2015) | 1.338 | 1.829 | 1.992 | 2.194 | 2.967 | 5.553 | 6.571 | 7.990 | | ConvLSTM Shi et al. (2015) | 2.759 | 2.787 | 2.948 | 12.882 | 12.272 | 14.920 |
| | Autoformer Wu et al. (2021b) | 1.258 | 1.982 | 2.682 | 2.695 | 2.578 | 4.598 | 6.395 | 6.087 | | Autoformer Wu et al. (2021b) | 2.798 | 2.878 | 3.212 | 12.489 | 12.976 | 15.345 |
| | SCINet Liu et al. (2022a) | 1.458 | 2.905 | 1.890 | 1.870 | 2.880 | 5.456 | 6.873 | 8.293 | | SCINet Liu et al. (2022a) | 2.902 | 2.789 | 3.404 | 12.213 | 13.643 | 15.895 |
| | FEDformer Zhou et al. (2022b) | 1.590 | 1.563 | 1.992 | 2.809 | 3.556 | 5.679 | 6.163 | 6.946 | | FEDformer Zhou et al. (2022b) | 2.709 | 2.778 | 3.112 | 11.678 | 13.234 | 15.543 |
| | Stationary Liu et al. (2022b) | 1.354 | 1.430 | 2.058 | 2.890 | 3.050 | 4.987 | 6.201 | 7.845 | | Stationary Liu et al. (2022b) | 2.765 | 2.987 | 3.641 | 12.975 | 12.075 | 14.887 |
| | RLinear Li et al. (2023c) | 1.673 | 1.256 | 1.890 | 2.450 | 2.990 | 4.678 | 5.987 | 6.289 | | RLinear Li et al. (2023c) | 2.834 | 3.543 | 3.342 | 11.897 | 12.675 | 15.967 |
| | PatchTST Li et al. (2023b) | 1.568 | 1.789 | 1.640 | 2.180 | 1.764 | 4.234 | 5.239 | 3.923 | | PatchTST Li et al. (2023b) | 2.623 | 3.234 | 3.375 | 12.456 | 11.907 | 15.325 |
| | Crossformer Zhang & Yan (2023b) | 1.456 | 1.590 | 1.678 | 1.990 | 1.678 | 3.989 | 4.786 | 5.257 | | Crossformer Zhang & Yan (2023b) | 2.854 | 2.878 | 3.123 | 12.654 | 12.985 | 15.564 |
| | TiDE Das et al. (2023b) | 1.555 | 1.728 | 1.430 | 1.789 | 2.278 | 3.278 | 3.987 | 4.890 | | TiDE Das et al. (2023b) | 2.542 | 2.690 | 3.078 | 12.267 | 12.754 | 14.243 |
| | TimesNet Wu et al. (2023) | 1.145 | 1.567 | 1.678 | 1.890 | 2.789 | 2.908 | 3.678 | 3.980 | | TimesNet Wu et al. (2023) | 2.876 | 2.879 | 3.321 | 11.654 | 11.754 | 14.675 |
| | DLinear Zeng et al. (2023b) | 1.134 | 1.556 | 1.567 | 1.567 | 1.890 | 2.465 | 2.967 | 3.653 | | DLinear Zeng et al. (2023b) | 2.567 | 3.365 | 3.145 | 11.687 | 11.946 | 13.990 |
| | iTransformer Nie et al. (2024) | 1.123 | 1.487 | 1.435 | 1.345 | 1.670 | 1.910 | 2.456 | 2.847 | | iTransformer Nie et al. (2024) | 2.680 | 2.989 | 3.076 | 10.456 | 11.896 | 13.696 |
| | **FATE (Ours)** | **1.021** | **1.217** | **1.346** | **1.131** | **1.464** | **1.660** | **1.844** | **2.238** | | **FATE (Ours)** | **2.174** | **2.665** | **2.695** | **8.515** | **10.914** | **13.523** |
| New York | Transformer Vaswani et al. (2017b) | 1.426 | 2.043 | 2.271 | 2.489 | 3.836 | 7.533 | 9.268 | 10.978 | Maastricht | Transformer Vaswani et al. (2017b) | 4.770 | 5.293 | 5.649 | 30.891 | 43.283 | 50.678 |
| | 3D CNN Mehrkanoon (2019a) | 1.835 | 2.316 | 2.833 | 2.673 | 5.587 | 9.159 | 13.468 | 11.964 | | 3D CNN Mehrkanoon (2019a) | 4.276 | 5.078 | 5.609 | 28.823 | 40.531 | 49.410 |
| | LSTM Hochreiter & Schmidhuber (1997) | 1.596 | 2.126 | 2.325 | 2.507 | 4.724 | 8.103 | 9.749 | 10.985 | | LSTM Hochreiter & Schmidhuber (1997) | 3.982 | 5.036 | 5.373 | 24.860 | 39.484 | 46.590 |
| | ConvLSTM Shi et al. (2015) | 2.394 | 2.134 | 2.419 | 2.104 | 4.949 | 7.790 | 9.257 | 10.341 | | ConvLSTM Shi et al. (2015) | 4.578 | 4.863 | 5.322 | 32.699 | 39.819 | 48.288 |
| | Autoformer Wu et al. (2021b) | 1.756 | 1.981 | 2.587 | 2.446 | 4.436 | 7.234 | 10.457 | 9.357 | | Autoformer Wu et al. (2021b) | 4.896 | 5.987 | 5.670 | 32.999 | 39.563 | 48.939 |
| | SCINet Liu et al. (2022a) | 1.940 | 1.879 | 2.859 | 2.976 | 4.876 | 6.905 | 12.755 | 10.345 | | SCINet Liu et al. (2022a) | 4.886 | 5.109 | 5.348 | 33.123 | 40.909 | 47.834 |
| | FEDformer Zhou et al. (2022b) | 1.650 | 1.809 | 2.865 | 2.768 | 4.345 | 6.469 | 12.657 | 9.235 | | FEDformer Zhou et al. (2022b) | 4.609 | 5.689 | 5.456 | 32.689 | 41.549 | 45.834 |
| | Stationary Liu et al. (2022b) | 1.903 | 1.980 | 2.786 | 2.567 | 3.957 | 7.458 | 11.466 | 9.587 | | Stationary Liu et al. (2022b) | 4.679 | 5.786 | 5.940 | 32.569 | 40.457 | 47.394 |
| | RLinear Li et al. (2023c) | 1.455 | 1.912 | 2.532 | 2.545 | 4.768 | 7.548 | 10.567 | 10.344 | | RLinear Li et al. (2023c) | 4.798 | 5.079 | 5.749 | 32.564 | 41.348 | 50.576 |
| | PatchTST Li et al. (2023b) | 1.465 | 1.893 | 2.230 | 2.443 | 4.534 | 5.990 | 13.565 | 10.497 | | PatchTST Li et al. (2023b) | 4.765 | 5.768 | 5.088 | 31.455 | 39.457 | 49.785 |
| | Crossformer Zhang & Yan (2023b) | 2.124 | 2.498 | 2.432 | 2.234 | 4.786 | 6.935 | 11.356 | 9.346 | | Crossformer Zhang & Yan (2023b) | 4.56 | 5.698 | 5.678 | 31.455 | 41.694 | 46.876 |
| | TiDE Das et al. (2023b) | 1.967 | 2.231 | 2.241 | 2.948 | 3.654 | 7.345 | 10.549 | 9.438 | | TiDE Das et al. (2023b) | 4.969 | 5.345 | 5.543 | 31.289 | 40.694 | 48.567 |
| | TimesNet Wu et al. (2023) | 1.567 | 1.890 | 2.532 | 2.468 | 3.234 | 7.095 | 9.657 | 10.348 | | TimesNet Wu et al. (2023) | 4.579 | 5.234 | 5.432 | 30.234 | 41.457 | 50.345 |
| | DLinear Zeng et al. (2023b) | 1.563 | 2.086 | 2.124 | 2.983 | 3.767 | 6.455 | 9.378 | 9.347 | | DLinear Zeng et al. (2023b) | 4.998 | 5.234 | 5.876 | 30.457 | 40.345 | 49.566 |
| | iTransformer Nie et al. (2024) | 1.274 | 1.908 | 2.343 | 2.435 | 3.555 | 5.839 | 7.994 | 8.904 | | iTransformer Nie et al. (2024) | 4.458 | 5.343 | 5.765 | 30.578 | 39.457 | 43.456 |
| | **FATE (Ours)** | **0.982** | **1.689** | **1.974** | **1.995** | **3.180** | **5.296** | **6.677** | **8.193** | | **FATE (Ours)** | **4.164** | **4.410** | **4.940** | **21.458** | **35.501** | **39.707** |
| Los Angeles | Transformer Vaswani et al. (2017b) | 1.426 | 2.043 | 2.271 | 2.489 | 3.836 | 7.533 | 9.268 | 10.978 | Munich | Transformer Vaswani et al. (2017b) | 4.136 | 5.286 | 5.275 | 23.954 | 39.057 | 43.526 |
| | 3D CNN Mehrkanoon (2019a) | 1.835 | 2.316 | 2.833 | 2.673 | 5.587 | 9.159 | 13.467 | 11.968 | | 3D CNN Mehrkanoon (2019a) | 3.931 | 5.049 | 5.262 | 24.870 | 39.578 | 43.507 |
| | LSTM Hochreiter & Schmidhuber (1997) | 1.296 | 2.026 | 2.325 | 2.207 | 4.724 | 8.403 | 9.749 | 10.983 | | LSTM Hochreiter & Schmidhuber (1997) | 3.551 | 4.730 | 5.189 | 20.235 | 34.021 | 42.733 |
| | ConvLSTM Shi et al. (2015) | 1.594 | 2.134 | 2.419 | 2.704 | 4.949 | 7.790 | 8.457 | 12.342 | | ConvLSTM Shi et al. (2015) | 3.974 | 4.830 | 5.023 | 22.484 | 35.401 | 37.767 |
| | Autoformer Wu et al. (2021b) | 1.645 | 2.457 | 2.856 | 2.980 | 3.886 | 9.203 | 13.124 | 12.588 | | Autoformer Wu et al. (2021b) | 3.958 | 5.890 | 5.456 | 22.467 | 37.347 | 40.458 |
| | SCINet Liu et al. (2022a) | 1.458 | 2.346 | 2.456 | 2.608 | 3.508 | 9.134 | 12.244 | 12.458 | | SCINet Liu et al. (2022a) | 4.545 | 5.461 | 5.546 | 20.567 | 40.890 | 44.102 |
| | FEDformer Zhou et al. (2022b) | 1.748 | 2.479 | 2.567 | 2.647 | 3.680 | 7.904 | 13.598 | 12.453 | | FEDformer Zhou et al. (2022b) | 3.957 | 5.563 | 5.986 | 23.467 | 39.834 | 42.549 |
| | Stationary Liu et al. (2022b) | 1.983 | 2.980 | 2.096 | 2.678 | 3.976 | 8.348 | 12.548 | 11.579 | | Stationary Liu et al. (2022b) | 3.589 | 5.970 | 5.446 | 22.366 | 38.787 | 43.124 |
| | RLinear Li et al. (2023c) | 1.849 | 2.228 | 2.345 | 2.956 | 3.578 | 8.438 | 11.959 | 11.345 | | RLinear Li et al. (2023c) | 3.335 | 5.348 | 5.785 | 20.456 | 39.456 | 42.693 |
| | PatchTST Li et al. (2023b) | 1.648 | 2.562 | 2.956 | 2.907 | 3.877 | 7.348 | 11.345 | 10.397 | | PatchTST Li et al. (2023b) | 3.595 | 5.795 | 5.679 | 21.458 | 40.683 | 42.458 |
| | Crossformer Zhang & Yan (2023b) | 1.843 | 2.875 | 2.645 | 2.845 | 3.689 | 6.937 | 10.543 | 10.458 | | Crossformer Zhang & Yan (2023b) | 3.579 | 5.675 | 5.685 | 23.546 | 40.348 | 44.939 |
| | TiDE Das et al. (2023b) | 1.937 | 2.780 | 2.454 | 2.689 | 3.273 | 6.348 | 9.434 | 10.439 | | TiDE Das et al. (2023b) | 3.584 | 5.235 | 5.436 | 23.754 | 39.457 | 44.345 |
| | TimesNet Wu et al. (2023) | 1.893 | 2.549 | 2.644 | 2.997 | 3.679 | 6.438 | 8.934 | 9.948 | | TimesNet Wu et al. (2023) | 3.545 | 5.344 | 5.543 | 23.456 | 38.458 | 42.589 |
| | DLinear Zeng et al. (2023b) | 1.457 | 2.456 | 2.344 | 2.578 | 3.879 | 6.349 | 8.282 | 9.348 | | DLinear Zeng et al. (2023b) | 3.565 | 5.234 | 5.567 | 22.546 | 37.459 | 43.548 |
| | iTransformer Nie et al. (2024) | 1.247 | 1.908 | 1.992 | 2.264 | 3.979 | 6.475 | 8.348 | 8.458 | | iTransformer Nie et al. (2024) | 3.234 | 4.948 | 5.745 | 21.455 | 36.845 | 40.347 |
| | **FATE (Ours)** | **1.183** | **1.530** | **1.920** | **2.041** | **3.180** | **5.496** | **6.677** | **8.185** | | **FATE (Ours)** | **3.196** | **4.335** | **4.925** | **19.927** | **32.454** | **36.309** |

**Continental-scale Forecasting.** On USA-Canada and Europe datasets, we evaluate 4–16 hour forecasts using MAE and MSE (Table 2). FATE consistently outperforms all baselines, including robust Transformers and linear models. For example, in Vancouver, FATE reduces MAE and MSE by up to 15.9% and 24.9%, respectively, over the best baseline. The Europe dataset exhibits similar trends, highlighting FATE's robustness and ability to model long-horizon temporal dynamics effectively.

**Large-Scale Spatiotemporal Forecasting.** On the LargeST dataset (Table 3), FATE achieves the lowest MAE and MSE (0.160 and 0.255), surpassing D2STGNN by 10.1% and 13.6%, respectively. These results demonstrate FATE's capacity to capture intricate spatiotemporal dependencies in large-scale traffic data, making it highly suitable for real-world forecasting applications.

**Benchmark Dataset Evaluation.** Across four standard multivariate time series datasets (ETTh1, ETTm2, Traffic, Weather5k; Table 4), FATE consistently achieves state-of-the-art or competitive performance. Notably: - ETTh1: 4.3% lower MAE, capturing fine-grained temporal patterns. - Traffic: compared to PatchTST, MAE is slightly higher, but MSE decreases by 3%. - Weather5k: 9.1% MAE and 12.3% MSE improvements over CI-TSMixer, demonstrating robustness

Table 3: Comparison of model performance on *LargeST* dataset. The best performing model is shown in **bold** and the second best in red for clarity.

| Model | MAE | MSE |
|---|---|---|
| LSTM Hochreiter & Schmidhuber (1997) | 0.266 | 0.417 |
| DRCNN Sun et al. (2021) | 0.213 | 0.333 |
| STNN Yin et al. (2021) | 0.186 | 0.311 |
| STGODE Fang et al. (2021) | 0.195 | 0.335 |
| DGCRN Li et al. (2023a) | 0.180 | 0.300 |
| D2STGNN Shao et al. (2022) | 0.178 | 0.295 |
| **FATE (Ours)** | **0.160** | **0.255** |

4 hours into the future  8 hours into the future  12 hours into the future  16 hours into the future

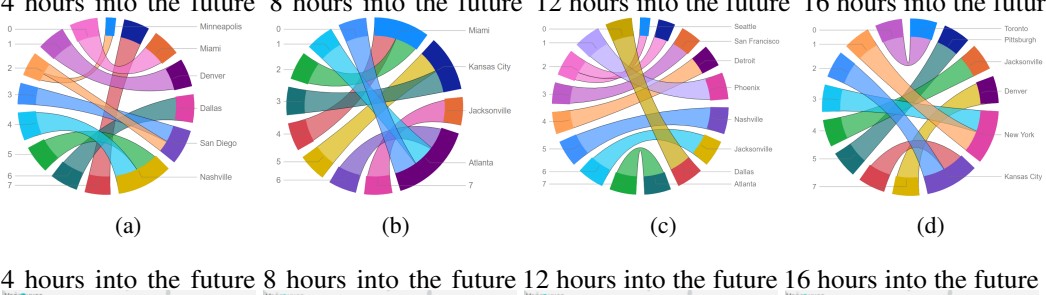

|  (a)  |  (b)  |  (c)  |  (d)  |

4 hours into the future  8 hours into the future  12 hours into the future  16 hours into the future

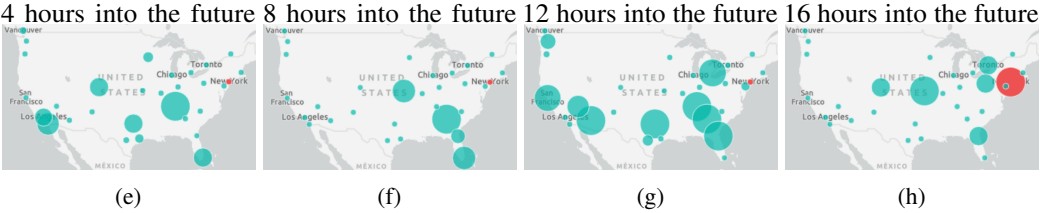

|  (e)  |  (f)  |  (g)  |  (h)  |

Figure 3: Attention visualization for **New York** in USA-Canada dataset. The circular graphs show which city each of the most important heads attends to. The thickness of the line represents the amount of attention each of the heads is paying to the cities. The size of the circles indicates the importance of Each city in the temperature prediction for the target city. The target city is marked as a red circle, and its size corresponds to the importance of the attention to itself.

to high-dimensional noise. - ETTm2: 13.3% MAE and 7.9% MSE improvements, confirming generalizability across diverse datasets. These results collectively validate FATE 's ability to model multi-scale temporal and spatial dependencies, yielding accurate and stable forecasts across both regional and large-scale datasets.

Table 4: Comparison of MAE and MSE on temperature prediction across diverse real-world multivariate time-series datasets. The best performing results are highlighted in **bold** and the second best are marked in red for clarity.

| **Model** | ETTH1 | | Traffic | | Weather5K | | ETTM2 | |
|---|---|---|---|---|---|---|---|---|
| | **MAE** | **MSE** | **MAE** | **MSE** | **MAE** | **MSE** | **MAE** | **MSE** |
| FATE (Ours) | **0.381** | 0.377 | 0.254 | **0.349** | **0.179** | **0.128** | **0.221** | **0.151** |
| CI-TSMixer Ekambaram et al. (2023) | 0.398 | **0.368** | 0.278 | 0.356 | 0.197 | 0.146 | 0.255 | 0.164 |
| PatchTST Li et al. (2023b) | 0.400 | 0.370 | **0.249** | 0.360 | 0.198 | 0.149 | 0.256 | 0.166 |
| DLinear Zeng et al. (2023b) | 0.399 | 0.375 | 0.282 | 0.410 | 0.237 | 0.176 | 0.260 | 0.167 |
| FEDformer Zhou et al. (2022b) | 0.419 | 0.376 | 0.366 | 0.587 | 0.296 | 0.217 | 0.287 | 0.203 |
| Autoformer Wu et al. (2021b) | 0.459 | 0.449 | 0.388 | 0.613 | 0.336 | 0.266 | 0.339 | 0.255 |
| Informer Zhou et al. (2021b) | 0.713 | 0.865 | 0.391 | 0.719 | 0.384 | 0.300 | 0.453 | 0.365 |

### 4.4 MODULATION VISUALIZATION AND ABLATION STUDY

Figure 3 illustrates the interpretability of FATE through focal modulation scores. Panels (a)–(d) show head-wise scores $N\widetilde{A}_s^h$ (Eq. 6), highlighting each head's focus in generating predictions. Aggregated city-wise scores $N\widetilde{A}_s$ (Eq. 7) reveal the contribution of each city to the target city. Panels (e)–(h) depict these interactions as graphs, where line thickness indicates attention strength and circle size represents city importance; the red circle marks the target city's self-attention. As forecast horizons extend, the target city increasingly attends to more distant contributors, reflecting dynamic spatiotemporal dependencies. Additional visualizations are provided in Appendix § A.1.

## 5 OUTLOOK AND FUTURE DIRECTIONS

The strong empirical performance of FATE opens multiple avenues for advancing spatio-temporal forecasting.

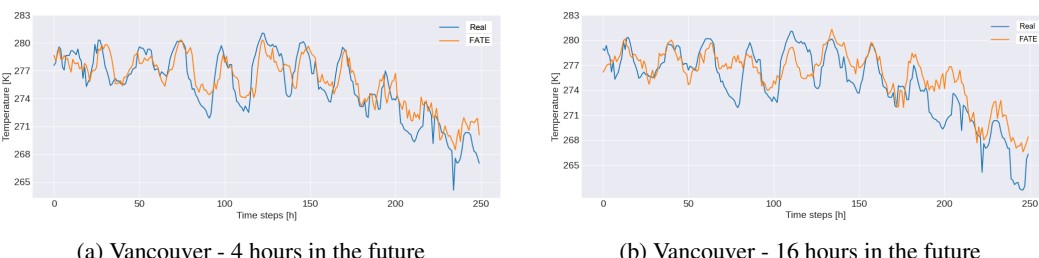

(a) Vancouver - 4 hours in the future        (b) Vancouver - 16 hours in the future

Figure 4: The comparison between the predictions of the FATE model and the real measurements for the hourly temperature of the test set of Vancouver.

**Scaling to global and ultra-long horizons.** While FATE performs strongly on regional datasets (Table 2), scaling to continental or global domains requires optimized training and inference. Future work may explore hierarchical or distributed focal-modulation architectures to retain interpretability while handling millions of spatial points over decades of data. **Richer variables and cross-domain fusion.** Current experiments emphasize temperature and standard meteorological features (Table 4). Adding variables such as precipitation, aerosols, oceanic indices, or soil moisture and fusing satellite imagery, reanalysis products, and socio-economic data could enhance predictive power and policy relevance. **Self-supervised pretraining.** Unlabeled climate data motivates self-supervised learning tailored to the focal-tensor setup. Objectives like contrastive or masked prediction can enrich spatio-temporal representations, improve robustness, and reduce dependence on labeled data. **Physics-informed inductive biases.** Incorporating physical constraints e.g., conservation laws or dynamical couplings into focal-modulation blocks may improve physical plausibility and reduce extrapolation error (Appendix §A.2). Hybrid integration with NWP ensembles is a promising future direction. **Efficiency and real-time inference.** Though efficient, FATE remains costlier than linear baselines. Techniques such as tensor compression, sparse kernels, or adaptive focal levels could enable lightweight, real-time variants for edge or on-device use. **Decision-support and societal impact.** Translating forecasts into actionable insights for agriculture, energy, and disaster response remains a key challenge. Interpretable modulation maps (Figure 3) and tailored visualizations can foster trust and support decision-making.

**Summary.** The tensorized focal-modulation design of FATE offers a scalable, extensible foundation for climate forecasting. Future extensions across scale, modality, physics, and application position it as a comprehensive tool for sustainable development.

## 6 CONCLUSION

In this study, we introduced the *Focal-Modulated Tensorized Encoder* (FATE), a framework designed to capture complex spatiotemporal dependencies in climate data. By leveraging tensorized focal modulation, FATE effectively models multi-scale interactions across time, space, and climate parameters. We evaluated FATE on seven diverse real-world multivariate time series datasets, consistently achieving state-of-the-art performance. Additionally, we proposed head-wise and city-wise modulation scores to enhance interpretability and conducted ablation studies to quantify their impact. This work provides a foundation for informed climate policy decisions and broader applications that exploit 3D tensor-structured data.

## LIMITATIONS

Our current evaluation focuses on temperature and related climate variables within mid-scale regional datasets. Extending FATE to additional meteorological variables and global-scale grids is a direction for future work. While FATE introduces modest computational overhead (trainable on a single A100 GPU), it remains practical for deployment and can be further optimized for edge or real-time applications. These limitations are operational rather than conceptual.

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

# A    APPENDIX

## A.1    FURTHER VISUALIZATIONS

We further visualized the feature selection process of the tensorial modulation mechanism, specifically focusing on the visualizations for selected cities. From a spatiotemporal perspective, the mechanism progressively emphasizes more distant cities as the prediction time step increases. This behavior highlights the model's ability to adaptively focus on relevant spatial regions over time.

The computations for the Query, Key, and Value tensors are defined as follows:

$$Q_{t,c,d} = X_{t,c,f} \cdot W^Q_{f,d,c} \quad \forall t = 1, \ldots, T, \quad c = 1, \ldots, C, \tag{8}$$

$$K_{t,c,d} = X_{t,c,f} \cdot W^K_{f,d,c} \quad \forall t = 1, \ldots, T, \quad c = 1, \ldots, C, \tag{9}$$

$$V_{t,c,d} = X_{t,c,f} \cdot W^V_{f,d,c} \quad \forall t = 1, \ldots, T, \quad c = 1, \ldots, C. \tag{10}$$

Here, $X_{t,c,f}$ represents the input tensor with temporal index $t$, spatial index $c$, and feature index $f$. The learnable weight matrices $W^Q_{f,d,c}$, $W^K_{f,d,c}$, and $W^V_{f,d,c}$ map the input features to the Query ($Q$), Key ($K$), and Value ($V$) tensors, respectively. These operations allow the model to dynamically compute across time, space, and features.

Maastricht - 2 days in the future   Maastricht - 4 days in the future   Maastricht - 6 days in the future

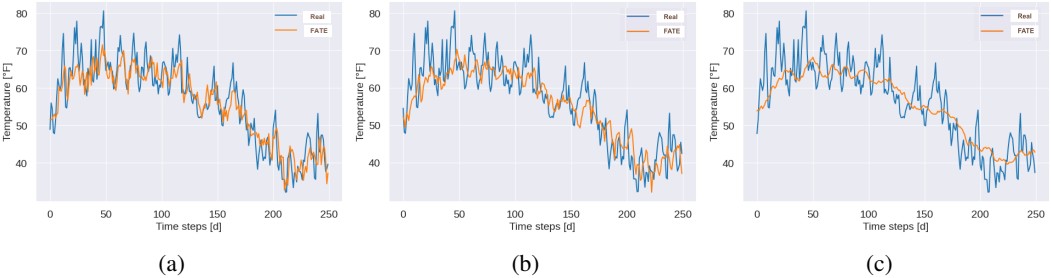

(a)                                (b)                                (c)

Figure 5: The comparison between the predictions of FATE model and the real measurements for **average daily temperature** of the test set of **Maastricht**.

Figure. 5 presents the model predictions alongside real measurements for Maastricht, showcasing 2, 4, and 6-day forecast horizons. While FATE accurately captures smaller variations for 2- and 4-day predictions, its performance over 6 days primarily reflects broader temperature trends. Unlike the previous dataset, the results on the Europe dataset demonstrate varying performance, with FATE ranking as the second-best model overall. Notably, FATE outperforms other models in predicting 4- and 6-day horizons specifically for Maastricht.

Experiments on this dataset were conducted for 2, 4, and 6 days ahead predictions, using an empirically determined lag parameter of 8 days to construct the regressors. Target cities included Barcelona,

Barcelona - 2 days in the future   Barcelona - 4 days in the future   Barcelona - 6 days in the future

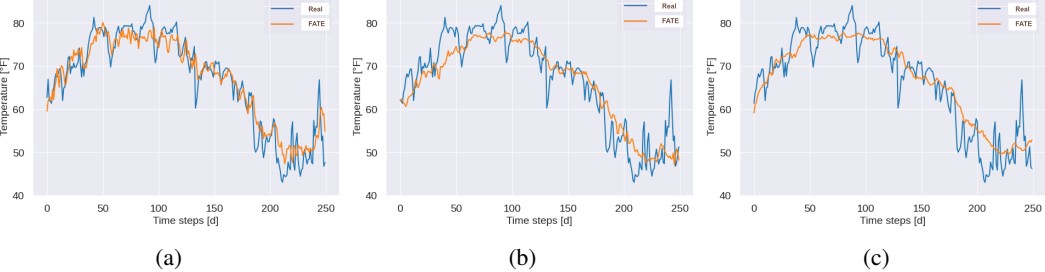

(a)                                (b)                                (c)

Figure 6: The comparison between the predictions of FATE model and the real measurements for **average daily temperature** of the test set of **Barcelona**.

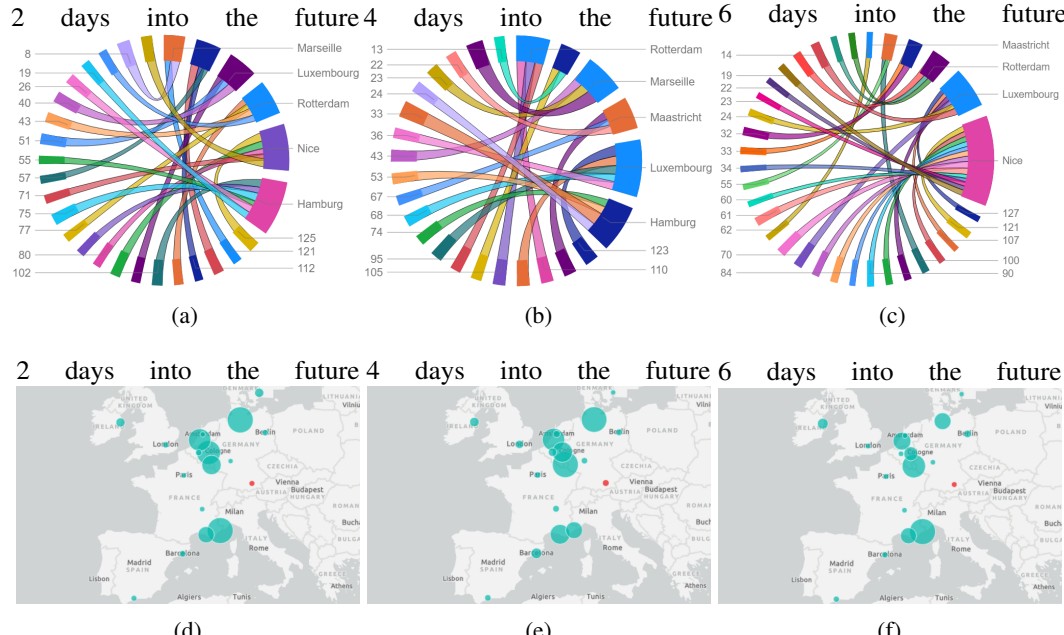

Figure 7: Focal Modulation visualization for **Munich** in Europe dataset. The top graphs show which city each of the heads attends. The thickness of the line represents the amount of modulation each of the heads is paying to the cities.

Maastricht, and Munich, with the average temperature as the primary prediction feature. Additionally, Figure. 6 highlights model predictions versus real measurements for Barcelona at 2, 4, and 6 days into the future. Despite FATE 's competitive performance in specific scenarios, the LSTM-based model achieved the lowest MAE in 5 city–time-step pairs and the lowest MSE in 4 pairs. Prior studies Guo et al. (2019); Ezen-Can (2020) have reported that Transformers can struggle in scenarios with limited data, which may explain why the Europe dataset constrained FATE 's performance compared to LSTM. Interestingly, in the Europe dataset, certain cities demonstrated minimal contribution to the predictions, suggesting inherent feature selection by the model. This observation is evident in Munich's predictions, shown in the Figure. 7, where the circular graphs and maps illustrate limited spatial dependencies for some cities. Unlike the US-Canada dataset, a distinct spatiotemporal pattern was not observed for Munich's predictions. Lastly, focal modulation visualizations are shown in the Figures. 8, 9, and 3 reveal both spatial and temporal dynamics, combining map-based views and circular graphs for each forecast horizon. These visualizations underline the adaptability of FATE in leveraging key features, particularly in datasets with varying data distributions and prediction horizons.

In this study, we leverage focal modulation weights to enhance model interpretability, specifically by identifying which areas of the input data the model prioritizes when making predictions. A major challenge in many practical applications is the cost of collecting labeled data, which often results in a limited number of training samples, particularly when dealing with high-dimensional datasets. This can lead to the curse of dimensionality, a significant hurdle when trying to effectively learn from such data. We focus on three primary challenges in this context. First, temperature forecasting is a multifaceted problem that requires not only past temperature data for the target location but also additional features such as wind speed, wind direction, atmospheric pressure, and humidity. These features add complexity to the model, making it crucial to handle high-dimensional data effectively. The second challenge arises from the increase in input dimensionality. This expansion must be reflected in the model's weight structure. One possible approach is to flatten the input data to preserve the transformer architecture as it is. However, this could lead to a loss of critical information, thereby degrading model performance. Alternatively, we could retain the full dimensionality of the input data, which would require expanding the model's capacity to handle this higher-dimensional space. While this method maintains data integrity, it also results in increased computational demands and

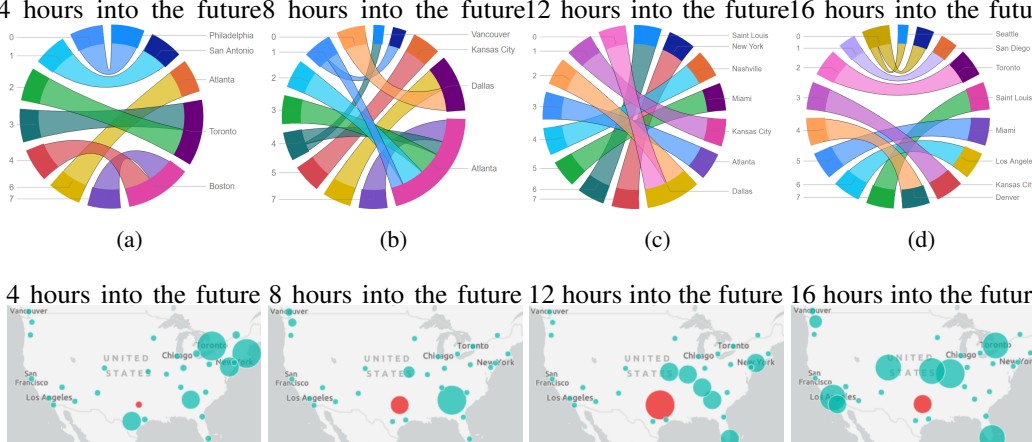

Figure 8: The Focal Modulation visualization for **Dallas** in the USA-Canada dataset illustrates the attention mechanism of the model. The circular graphs depict which cities are attended to by the most important attention heads. The line thickness represents the strength of the attention each head allocates to these cities, while the circle size indicates the relative importance of each city in predicting the temperature for the target city. The target city, marked by a red circle, has its size proportional to the level of focal modulation it receives from the model.

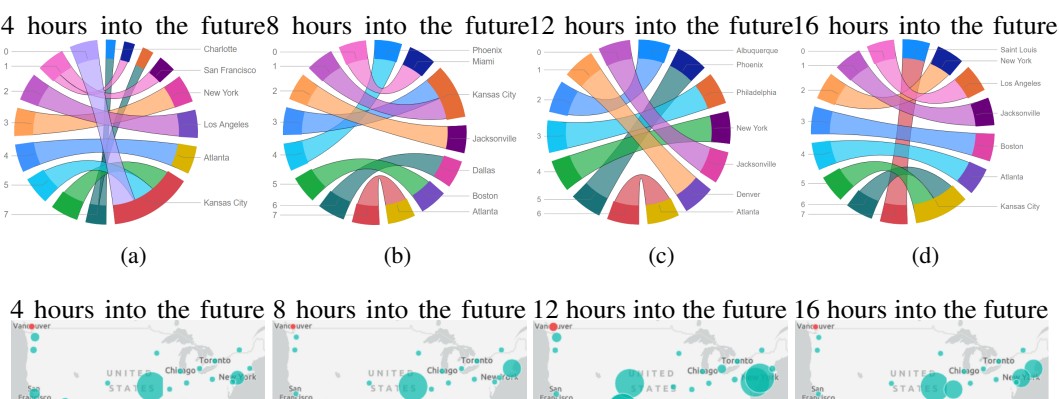

Figure 9: Focal Modulation visualization for **Vancouver** in USA-Canada dataset. The circular graphs show which city each of the most important heads attends to. The thickness of the line represents the amount of attention each of the heads is paying to the cities. The size of the circles indicates the importance of Each city in the temperature prediction for the target city. The target city is marked as a red circle, and its size corresponds to the importance of the focal modulation to itself.

longer training times. A potential solution to this issue is the use of Tensor Processing Units (TPUs), which can significantly speed up both training and evaluation phases. The third challenge is related to model explainability, which has become a pressing concern as models are increasingly used to automate tasks without transparent reasoning behind their predictions. To address this, we utilize focal modulation weights to pinpoint the areas of the input that the model focuses on most heavily when making its predictions, thereby offering valuable insights into its decision-making process. By tackling these challenges, this work contributes to improving both the efficiency and interpretability of temperature forecasting models in high-dimensional settings.

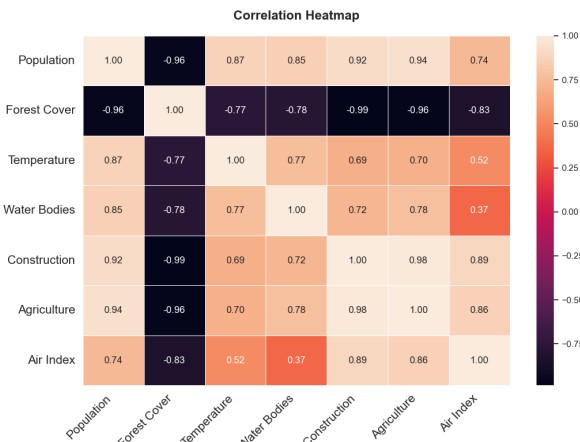

Figure 10: Correlation Heatmap: It illustrates the relationships between various climatic parameters

## A.2 CORRELATION BETWEEN PARAMETERS

The correlation matrix, as depicted in Figure 10, provides a comprehensive analysis of the relationships between the seven selected parameters: Air Index, Forest Cover, Water Bodies, Agriculture and Vegetation, Population, Surface Temperature, and Construction. The matrix reveals the intricate interdependencies among these variables, offering insight into the underlying dynamics of the study area. Notably, most parameters exhibit positive correlations, suggesting that as one variable increases, others tend to follow suit. For example, it is expected that an increase in population may lead to higher construction activity and possibly a reduction in forest cover. Similarly, an increase in agricultural and vegetation areas may correlate with changes in surface temperature or water body extent.

In contrast, the parameter *Forest Cover* stands out due to its negative correlation with several other parameters. The gradual reduction in forest cover over time reflects the increasing anthropogenic activities such as construction and agriculture. This negative correlation is indicative of environmental degradation, as the expansion of urban areas and agricultural practices leads to deforestation, which in turn impacts other environmental factors. The relationship between these variables underscores the complexity of the region's ecological balance and emphasizes the need for sustainable practices to mitigate the adverse effects of rapid development.

The correlation matrix serves as a vital tool in understanding the interconnectedness of these environmental and socio-economic parameters, guiding future analyses and policy recommendations aimed at fostering more sustainable development strategies.

