# OpenReview forum: "FATE: Focal-modulated Attention Encoder for Multivariate Time-series Forecasting"
_ICLR.cc/2026/Conference — ICLR 2026 Conference Withdrawn Submission_

### Official Review · Reviewer_8UPm · 2025-10-16

**Soundness:** 2
**Presentation:** 1
**Contribution:** 2
**Rating:** 2
**Confidence:** 3

**Summary:**

This paper proposes FATE, which focuses on capturing spatial-temporal dependency by extending the prior work, FocalNet Transformer, to time series forecasting.

**Strengths:**

The exploration of spatial-temporal data with the concept of locality is plausible.

The experiment expands on multiple areas to show the model's performance gains.

**Weaknesses:**

**W1:** Notation lacks definitions: The notation system can be improved. A lot of terms have been used before defining. For example, in lines 51-53, the authors directly used T, S, and P for the 3-dimensional tensor without explaining what each dimension stands for. In Eq. (1) PE(\cdot) is also not explained, and many more. Additionally, many notations are not in math format. While these issues do not directly obscure the main ideas, they reduce the clarity and readability of the paper.

**W2:** Terms unexplained: A lot of uncommon terms are used without justification. For example, what are focal levels?

**W3:** Notations need to be revised: Apart from unexplained notations, the dimension of the tensors and weights also seems to be problematic. For example, in Eq. (2), it is unexplained how every h in Q, K, V can be obtained by multiplying every p in X with the corresponding weights. The dimension is either mismatched or requires a more detailed explanation. This paper needs great work on improving the notation clarity.

**W4:** Experiment not explained: The method was proposed for spatial-temporal data. However, in the tasks that the authors are using to evaluate the performance, many are not spatial-temporal data and thus do not have a "3-dimensional" tensor structure or any explicit geographical relationships (e.g., ETTh1, ETTm2, and Traffic). It generally does not make sense as the "city" or "local" concept becomes invalid for these specific datasets, and there is also no explanation of how to adapt the proposed model to these tasks that hold unordered covariates.

**W5:** Insufficient analysis. The paper would benefit from additional experiments to provide a deeper understanding of the model’s behaviour. For example, several parameters specific to this work are presented in Table 1 without validation or discussion of their impact.


Minor:

- The used baselines are quite out-of-date, with the latest being iTransformer, which was published in 2024. Many prior work have had lower MAE scores on the benchmark dataset than this work.

- Color scale can be improved. The current version is not color-blind friendly.

**Questions:**

- Please explain in detail what are the 4 focal levels for temporal sequences?

---

### Official Review · Reviewer_X1xV · 2025-10-30

**Soundness:** 3
**Presentation:** 3
**Contribution:** 2
**Rating:** 4
**Confidence:** 4

**Summary:**

The paper addresses key challenges in multivariate time-series forecasting (MTSF), including the difficulty of capturing hierarchical spatiotemporal dependencies and computational inefficiencies in high-dimensional data. To tackle these issues, the authors propose **FATE (Focal-modulated Attention Encoder)**, a novel Transformer-based architecture designed to improve accuracy and scalability for long-horizon and high-dimensional forecasting tasks.

Key contributions include:
1. **Tensorized Focal Modulation**: Retains the full 3D structure of the input tensor (temporal, spatial, and feature dimensions), enabling effective modeling of long-range dependencies.
2. **Focal Temporal Grouping**: Dynamically defines temporal focal groups to capture hierarchical temporal dependencies.
3. **Cross-axis Modulation**: Extends focal modulation to the feature dimension to model cross-feature interactions.

The authors validate FATE on seven diverse datasets, demonstrating state-of-the-art (SOTA) performance, particularly on long-horizon and high-variability tasks. Extensive ablations and interpretability visualizations further support the effectiveness of the proposed method.

**Strengths:**

- **Innovative Architectural Design**
  The use of tensorized focal modulation to preserve the 3D input structure is a significant innovation. This approach effectively models spatiotemporal dependencies and cross-feature interactions, which are critical for multivariate time-series forecasting.

- **Thorough Evaluation**
  The authors conduct comprehensive experiments, including ablation studies, to validate the impact of each component. Visualizations of focal modulation and attention dynamics further strengthen the empirical claims.

**Weaknesses:**

- **Limited Discussion on Computational Trade-offs**
  While the paper highlights that FATE introduces moderate computational overhead, it does not provide a detailed comparison of training and inference times against lightweight baselines like linear models.

- **Inconsistent Performance on Certain Datasets**
  Although FATE achieves SOTA results on most benchmarks, its performance on the Europe dataset is relatively weaker, with LSTM models outperforming it in several scenarios.

- **Sparse Explanation of Feature Selection**
  The paper mentions the use of seven meteorological features for climate datasets but does not elaborate on the rationale or methodology for feature selection.

**Questions:**

1. **Computational Efficiency**
   Could the authors provide more detailed runtime comparisons (e.g., training/inference time per epoch) between FATE and lightweight baselines (e.g., DLinear, TiDE)?

2. **Generalization to Limited Data**
   The Europe dataset results suggest that FATE may struggle in scenarios with limited or unevenly distributed data. Could the authors discuss strategies to improve FATE's robustness in such cases?

3. **Feature Selection**
   How were the seven meteorological features chosen for the climate datasets? Did the authors perform any feature importance analysis to validate their selection?

**Details Of Ethics Concerns:**

1. **Computational Overhead**
   Despite its modular efficiency, FATE introduces additional computational complexity compared to lightweight baselines, which may limit its deployment in edge or real-time applications.

2. **Weaker Performance on Limited Data**
   FATE's performance is inconsistent on datasets with limited or heterogeneous distributions (e.g., Europe), indicating potential limitations in generalization.

3. **Feature Selection Process**
   The lack of a detailed feature selection methodology reduces reproducibility and raises questions about the robustness of the results across different domains.

---

### Official Review · Reviewer_KfDh · 2025-11-01

**Soundness:** 3
**Presentation:** 3
**Contribution:** 2
**Rating:** 4
**Confidence:** 3

**Summary:**

The paper proposes FATE (Focal-modulated Attention Encoder), an encoder-only Transformer for multivariate time-series forecasting. The key ideas are: (i) tensorized QKV projections that preserve the native 3D structure of inputs; (ii) a tensorial focal-modulation mechanism that aggregates hierarchical temporal context while gating interactions across stations/variables; and (iii) dual modulation scores to provide interpretability about which sources drive forecasts. Experiments on seven datasets show consistent gains against Transformer, linear, and spatio-temporal GNN baselines, with ablations and qualitative visualizations supporting the design.

**Strengths:**

1. This paper extends focal modulation (from vision) to a tensorized, dual-axis scheme for multivariate time series, preserving temporal and variable axes and introducing dual modulation scores for interpretability.
2. Broad evaluation across 7 datasets with long-horizon regimes is conducted; consistent accuracy gains are reported, including on large-scale traffic where FATE improves MAE/MSE over the best GNN baselines. Qualitative modulation maps align with the narrative about dynamic spatial dependencies.
3. Figures explaining slice-wise QKV formation and multi-head aggregation improve intuition, and the modulation visualizations make the interpretability story tangible.
4. The approach targets real forecasting challenges and shows benefits at longer horizons—valuable for climate and infrastructure planning.

**Weaknesses:**

1. The distinctions from recent tensor/patch or efficiency-oriented methods (e.g., tensorized attention variants, multi-scale mixers) are not clear; ablations isolate focal levels and gating qualitatively, but do not study which tensorization choices (e.g., per-axis PE, grouped projections) are essential vs. incidental. A one-for-one replacement study (e.g., FATE vs. Time-tensorized attention with identical training) is missing.
2. The paper states “moderate overhead” and “comparable to baseline Transformers,” but provides limited wall-clock/GPU memory curves across sequence length, station count, and horizon. More rigorous asymptotic and empirical scaling (vs. quadratic attention and linear baselines) would strengthen soundness.
3. While code is provided, some training details remain high-level (e.g., feature preprocessing per dataset; normalization and leakage controls; early stopping; number of runs). HP tables are present but do not specify search ranges or budget comparability across models.
4. Modulation scores are visually appealing, yet the paper does not quantify faithfulness (e.g., perturbation tests, deletion/insertion curves) or compare to attention-based or attribution baselines.

**Questions:**

1. Please include runtime and peak memory vs. (i) number of stations, (ii) horizon length, and (iii) input window, comparing to vanilla Transformers and linear baselines on identical hardware/settings.
2. How many random seeds per result? Could you report mean and std and significance tests for Tables 2–4? Also, are hyperparameter budgets matched across baselines?
3. Beyond the circular graphs, can you quantify whether high-score cities/heads are causally important (e.g., occlusion/ablation of stations or time segments reduces accuracy proportionally to scores)?

---

### Official Review · Reviewer_Lf8w · 2025-11-01

**Soundness:** 3
**Presentation:** 3
**Contribution:** 3
**Rating:** 6
**Confidence:** 3

**Summary:**

The paper proposes FATE, a Focal-Modulated Tensorized Encoder tailored for multivariate time-series forecasting. Unlike vanilla Transformers that flatten inputs, FATE preserves a 3D tensor structure, introduces tensorized focal modulation with temporal focal grouping and cross-axis (feature) modulation, and provides interpretability via dual modulation scores (head-wise and station-wise). Across seven datasets, FATE reports consistent SOTA or competitive performance, with notable gains on long horizons and high-dimensional settings. Ablations, visualizations, and limited efficiency analysis are included.

**Strengths:**

1 Clear architectural novelty: a principled tensorized design that preserves temporal and feature axes; focal modulation tailored to time-series (temporal focal groups) rather than spatial grids; cross-axis modulation for multivariate dependencies.

2 Interpretability: dual modulation scores linking heads to stations, with compelling visualizations that show evolving spatial focus as horizon increases.

3 Strong empirical performance: consistent improvements on diverse datasets, including large-scale (LargeST) and climate-related (Weather5k), and competitive performance on standard ETT benchmarks. The reported margins on several tasks are sizable.

**Weaknesses:**

1 Hyperparameters differ substantially across models (batch sizes, layers/heads), and several strong modern baselines are missing or lightly tuned

2 Claims of “moderate overhead” vs. Transformers are not backed by FLOPs/latency/memory scaling curves across sequence length, stations, and features; no wall-clock comparisons at different horizons or ablations on focal levels.

3 For Traffic, FATE’s MAE is slightly worse than PatchTST (though MSE improves). A broader analysis of error distributions would clarify the trade-offs.

**Questions:**

1 Can you report FLOPs, max memory, and throughput vs. (T, S, P) compared to standard Transformer, PatchTST, and TimeTensor, across short and long horizons?

---

### Note · Authors · 2025-11-12

**Comment:**

We, the authors, wish to withdraw our submission “FATE: Focal-modulated Attention Encoder for Multivariate Time-series Forecasting” from consideration at ICLR 2026, as the current version requires further improvements and additional validation before resubmission.

**Withdrawal Confirmation:**

I have read and agree with the venue's withdrawal policy on behalf of myself and my co-authors.